# Non-alcoholic fatty liver disease is not associated with impairment in health-related quality of life in virally suppressed persons with human immune deficiency virus

**Samer Gawrieh**[1]*, **Kathleen E. Corey**[2], **Jordan E. Lake**[3], **Niharika Samala**[1], **Archita P. Desai**[1], **Paula Debroy**[3], **Julia A. Sjoquist**[2], **Montreca Robison**[1], **Mark Tann**[4], **Fatih Akisik**[4], **Surya S. Bhamidipalli**[5], **Chandan K. Saha**[5], **Kimon Zachary**[6], **Gregory K. Robbins**[6], **Samir K. Gupta**[7], **Raymond T. Chung**[2], **Naga Chalasani**[1]

1 Division of Gastroenterology and Hepatology, Department of Medicine, Indiana University School of Medicine, Indianapolis, Indiana, United Sates of America, 2 Liver Center, Division of Gastroenterology, Department of Medicine, Massachusetts General Hospital, Harvard Medical School, Boston, Massachusetts, United States of America, 3 Division of Infectious Diseases, Department of Medicine, University of Texas Health Science Center at Houston, Houston, Texas, United States of America, 4 Department of Radiology, Indiana University School of Medicine, Indianapolis, Indiana, United States of America, 5 Department of Biostatistics and Health Data Science, Indiana University School of Medicine, Indianapolis, Indiana, United States of America, 6 Division of Infectious Diseases, Department of Medicine, Massachusetts General Hospital, Harvard Medical School, Boston, Massachusetts, United States of America, 7 Division of Infectious Diseases, Department of Medicine, Indiana University School of Medicine, Indianapolis, Indiana, United States of America

* sgawrieh@iu.edu

**Data Availability Statement:** All relevant data are within the paper and its Supporting Information file.

## Abstract

Non-alcoholic fatty liver disease (NAFLD) is the most common liver disease in persons with HIV (PWH) (HIV-NAFLD). It is unknown if HIV-NAFLD is associated with impairment in health-related quality of life (HRQOL). We examined HRQOL in PWH with and without NAFLD, compared HRQOL in HIV- versus primary NAFLD, and determined factors associated with HRQOL in these groups. Prospectively enrolled 200 PWH and 474 participants with primary NAFLD completed the Rand SF-36 assessment which measures 8 domains of HRQOL. Individual domain scores were used to create composite physical and mental component summary scores. Univariate and multivariate analyses determined variables associated with HRQOL in PWH and in HIV- and primary NAFLD. In PWH, 48% had HIV-NAFLD, 10.2% had clinically significant fibrosis, 99.5% were on antiretroviral therapy, and 96.5% had HIV RNA <200 copies/ml. There was no difference in HRQOL in PWH with or without NAFLD. Diabetes, non-Hispanic ethnicity, and nadir CD4 counts were independently associated with impaired HRQOL in PWH. In HIV-NAFLD, HRQOL did not differ between participants with or without clinically significant fibrosis. Participants with HIV-NAFLD compared to those with primary NAFLD were less frequently cisgender females, White, more frequently Hispanic, had lower BMI and lower frequency of obesity and diabetes. HRQOL of individuals with HIV-NAFLD was not significantly different from those with primary NAFLD. In conclusion, in virally suppressed

**Funding:** NIDDK R01 DK112293 to SG, R01 DK126042 to JEL. The funders had no role in study design, data collection and analysis, decision to publish, or preparation of the manuscript.

**Competing interests:** Dr. Gawrieh consulting: TransMedics, Pfizer. Research grant support: Cirius, Galmed, Viking and Zydus. Dr. Corey serves on the scientific advisory board for Theratechnologies, Novo Nordisk and BMS and has received grant funding from Boehringer-Ingelheim, BMS and Novartis, Dr. Lake receives research support from Gilead Sciences and Zydus. In the past 12 months she has received research support from Pfizer, CytoDyn, and Oncoimmune, and has served as a consultant to Merck and Theratechnologies, Dr. Samala, Dr. Desai, Dr. Debroy, Ms.Sjoquist, Ms. Robison, Ms. Bhamidipalli, Dr. Saha, Dr. Zachary, Dr. Akisik, and Dr. Tann have nothing to disclose. Dr. Robbin has served as a consultant for SEED , Dr. Gupta reports consultancy/advisory fees from Gilead Sciences, Inc. and ViiV Healthcare and research support from the NIH, Indiana University School of Medicine, and ViiV HealthCare, Dr. Chung has received research grant support (to institution) from Boehringer Ingelheim, BMS, Roche, Gilead, Janssen, and GSK , Dr. Chalasani has ongoing research support from Eli Lilly, Galectin Therapeutics, Intercept, and Exact Sciences, In the past 12 months, he has received consulting fees from Abbvie, Madrigal, Nusirt, Allergan, Siemens, Genentech, Zydus, La Jolla, Axcella, Foresite Labs, and Galectin Therapeutics. This does not alter our adherence to PLOS ONE policies on sharing data and materials.

PWH, HRQOL is not different between participants with or without HIV-NAFLD. HRQOL is not different between HIV-NAFLD and primary NAFLD.

## Introduction

Non-alcoholic fatty liver disease (NAFLD) is the most common liver disease in persons without human immune deficiency virus (HIV) (primary NAFLD), affecting nearly a quarter of the population in North America [1]. Primary NAFLD has a spectrum that starts with simple accumulation of triglycerides in the liver and extends to a more severe and progressive phenotype, non-alcoholic steatohepatitis (NASH), where in addition to steatosis, there is liver cell injury, inflammation, and fibrosis [2, 3]. Patients with NAFLD and especially those with hepatic fibrosis are at increased risk for liver-related outcomes and mortality [4–6]. NAFLD is currently a leading cause of end-stage liver disease and hepatocellular carcinoma, and the most rapidly rising indication for liver transplantation in the US [7, 8].

The increasing efficacy and utilization of antiretroviral therapy (ART) has improved the longevity of persons with HIV (PWH) [9, 10]. In the era of effective ART, PWH experience ART-associated weight gain and lipodystrophy, and increased morbidity and mortality from non-AIDS-related illnesses such as liver, metabolic and cardiovascular diseases [11–14]. NAFLD has emerged as the most common liver disease in this population (HIV NAFLD) [15–17].

In addition to its impact on morbidity and mortality, NAFLD burden extends to affect health-related quality of life (HRQOL), as evaluated by a person's perception of the physical, mental and emotional aspects of their well-being. Studies show persons with primary NAFLD have significant reduction in HRQOL compared to the general population [18–20]. The impairment is more pronounced in the physical domains of HRQOL and worse in NAFLD patients with NASH, advanced fibrosis or cirrhosis [19, 21–23]. Despite significant improvement in survival on ART, virally suppressed PWH still report HRQOL that is worse than the general population [24, 25]. Whether HIV-NAFLD further impacts the HRQOL in virally suppressed, HIV mono-infected persons is unknown. Further, how HRQOL of PWH and NAFLD compares to that of persons with primary NAFLD has not been examined.

In this study, we first assessed HRQOL in a well phenotyped cohort of PWH with and without HIV NAFLD. We next compared HRQOL in persons with HIV-NAFLD to that of a large cohort of well characterized adults with primary NAFLD. Finally, we sought to determine factors associated with the physical and mental components of HRQOL in PWH and in NAFLD.

## Methods

### Identification of study participants

HIV-NAFLD cohort: Consecutive consenting PWH were prospectively enrolled from three outpatient HIV clinics at Indiana University School of Medicine, Massachusetts General Hospital and University of Texas Health Science Center between 2018 and 2022. Each participant provided a signed informed consent. The study protocol was approved by each site's Institutional Review Board (IRB). Inclusion criteria were hepatic steatosis by ultrasound [3], age $\geq$ 18 years, documented HIV defined by a positive HIV antibody assay and/or detectable HIV-1 RNA, and stable ART regimen for three months prior to enrollment. Exclusion criteria were excessive alcohol use defined by Alcohol Use Disorders Identification Test (AUDIT) score of $\geq$8, evidence of hepatitis B or C, or known other liver disease such as autoimmune hepatitis, cholestatic liver diseases, Wilson disease, hemochromatosis, etc.

Primary NAFLD cohort: Consecutive consenting patients without HIV were prospectively enrolled from the NAFLD clinic at Indiana University School of Medicine between 2017–2022. Indiana University IRB approved the protocol. Each participant provided a signed informed consent. NAFLD was diagnosed based on the recent American Association for the Study of Liver Diseases guidelines [3], which requires the presence of hepatic steatosis, either by imaging or histology, absence of other liver diseases, and lack of secondary causes of hepatic fat accumulation such as significant alcohol consumption, long-term use of a steatogenic medication, or monogenic hereditary disorders. Each participant provided a signed informed consent.

## Characterization of study participants

For PWH, a trained study physician or technician performed liver imaging with ultrasound, which was centrally read by two experienced radiologists to determine the presence of fatty liver (steatosis). Participants then underwent vibration controlled transient elastography (VCTE) by Fibroscan® to obtain controlled attenuation parameter (CAP) and liver stiffness measurement (LSM). Each participant underwent history and physical examination by a study physician. Extensive data were collected including demographic (age; sex; race; ethnicity), anthropometrics [body mass index (BMI), waist circumference], vital signs, medical and medicinal history, HIV and ART data (HIV RNA load, CD4$^+$ T cell nadir and current count at enrollment, current and prior ART classes). Clinically significant fibrosis was defined as LSM$\geq$ 8.6 kPa [26].

For patients with primary NAFLD, demographic, anthropometric, clinical, VCTE, and laboratory data were systematically collected on all participants.

## Study questionnaires

Each subject completed an AUDIT questionnaire to assess alcohol consumption in the past year. AUDIT is a simple ten-question test developed by the World Health Organization to determine if a person is a risky drinker or has alcohol use disorders. This instrument has been validated and allows quantification of daily drinks of alcohol consumed. A score of 8 or more indicates a strong likelihood of risky drinking or harmful alcohol use [27, 28].

At the time of enrollment (same day of the diagnostic testing for NAFLD), participants also completed the 36-item Short Form (SF-36) Health Survey version 1.0. SF-36 is a widely used and validated tool to evaluate the impact of disease on several physical and mental domains of health in the setting of a variety of chronic diseases, including liver diseases, primary NAFLD and HIV [29–33]. SF-36 measures HRQOL in 8 dimensions: physical functioning, role limitations due to physical health, emotional well-being, role limitations due to emotional problems, energy/fatigue, social functioning, pain, and general health. Each scale of the SF-36 was transformed to a continuous scale ranging between 0 and 100, and scores were calibrated in such a way that 50 is the average, with higher scores reflecting better health. The SF-36 dimensions were summarized using the physical component summary score (PCS) and mental component summary score (MCS) as previously described [34, 35]. In the US general population, the mean for the PCS and MCS is 50 with a standard deviation of 10 [35], thus a score more than 50 indicates better health and a score < 50 indicates poorer health than the US general population.

## Statistical methods

Participants with HIV-NAFLD versus without HIV-NAFLD as well as participants with HIV-NAFLD with versus without clinically significant fibrosis were compared in socio-demographic, laboratory and clinical variables. Means and standard deviations for continuous variables and

frequencies and percentages for categorical variables were provided for each of the four groups. Two-group comparisons were made using two-sample independent t-tests for continuous variables and Chi-square/Fishers test for categorical variables. Similar analyses were also performed for PCS and MCS. One of our primary aims was to identify factors associated with HRQOL in terms of PCS and MCS. We first assessed univariate or unadjusted association with quality of life and then we built a multivariate regression model using stepwise selection procedure.

A secondary aim was to assess how HIV-NAFLD versus primary NAFLD was differentially associated with HRQOL in terms of PCS and MCS after adjustment for differences in HIV-NAFLD and primary NAFLD groups. Although we could have considered a matched case (HIV NAFLD) control (primary NAFLD) study design by matching age and sex, we could not make the two groups similar in terms of other important covariates that were significantly different between the two groups. Therefore, we used multivariate analysis to adjust for all covariates which showed significantly different distribution between the two groups. This approach provided the advantage of maximizing the number of cases and controls we could use in statistical analyses. A p-value of 0.05 or less was considered statistically significant. SAS 9.4 (Cary, NC) was the software used for the analysis.

## Results

### Characteristics of the PWH cohort

A total of 200 PWH were evaluated in this analysis. Mean (SD) age was 50 (12.1) years, BMI 29.1 (6.0) kg/m$^2$, 73% were males, 56.5% White, and 30% Hispanic Table 1. All but one participant (99.5%) was on any form of ART and 96.5% had HIV RNA <200 copies/ml. Nearly half (48%, 96/200) of the patients had HIV-NAFLD and 10.2% (20/195) had clinically significant fibrosis.

PWH with HIV NAFLD, compared to PWH without HIV NAFLD, had higher mean (SD) BMI [30.5 (5.7) vs 27.7 (6.0) kg/m$^2$], were more likely to be White (69.8% vs 44.2%), had larger waist circumference [105.7 (15.6) vs 97.3 (15.1) cm], higher ALT [37.4 (24.3) vs 22.8 (11.8) U/L], and higher AST [30.0 (16.8) vs 21.3 (7.9) U/L] (p<0.01 for all). They also tended to have higher frequency of type 2 diabetes (16.8% vs 7.7%, p = 0.05) and use of hypoglycemic agents (16.8% vs 7.7%, p = 0.05). There were no differences between the two groups in age, proportion with history of acquired immunodeficiency syndrome, absolute or nadir CD4$^+$ T cell counts, proportion with nadir CD4 <200, ART exposure, or HIV-1 RNA suppression levels Table 1. PWH with HIV NAFLD, compared to PWH without HIV NAFLD tended to have longer duration of HIV infection [16.6 (10.1) vs 14.0 (9.6) years, P = 0.07], had more exposure to non-nucleoside reverse transcriptase inhibitors (20.8% vs 11.5%, P = 0.05) and Tenofovir Alafenamide (72.4% vs 59%, P = 0.06). The characteristics of PWH and clinically significant fibrosis are shown in Table 1.

### HRQOL in PWH

Overall, PWH had poor HRQOL as indicated by the low (≤50) mean (SD) PCS [47.7 (11.0)] and MCS [50.3 (12.2)] Table 2. No differences were detected in PCS, MCS or their subcomponents between PWH without and with HIV-NAFLD Table 2.

In subgroup analysis of HIV NAFLD, no significant differences were observed between those with and without clinically significant fibrosis Table 2.

### Factors associated with HRQOL in PWH

On univariate analysis (**S1 Table**), NAFLD or NAFLD with clinically significant fibrosis were not associated with PCS or MCS in PWH. Older age, higher BMI, Black race, larger waist circumference, diabetes and higher triglycerides levels were associated with worse PCS. Black

**Table 1. Characteristics of persons with HIV.**

| Variable | Total (N = 200) | No NAFLD (N = 104) | NAFLD (N = 96) | P-value | No CSF* (N = 175) | CSF (N = 20) | P-value |
|---|---|---|---|---|---|---|---|
| Age | 50.0 (12.1) | 49.2 (13.3) | 51.0 (10.7) | 0.28 | 49.6 (12.2) | 52.1 (11.1) | 0.38 |
| BMI (kg/m$^2$) | 29.1 (6.0) | 27.7 (6.0) | 30.5 (5.7) | <0.01 | 28.9 (5.9) | 30.9 (5.7) | 0.15 |
| Gender | | | | 0.22 | | | 1.00 |
| Cisgender Male | 146 (73%) | 75 (72.1%) | 71 (74.0%) | | 126 (72%) | 15 (75%) | |
| Cisgender Female | 41 (20.5%) | 25 (24.0%) | 16 (16.7%) | | 37 (21.1%) | 4 (20%) | |
| Transgender female | 12 (6%) | 4 (3.9%) | 8 (8.3%) | | 11 (6.3%) | 1 (5%) | |
| Other | 1 (0.5%) | 0 | 1 (1%) | | 1 (0.6%) | 0 | |
| Race | | | | <0.01 | | | 0.56 |
| White | 113 (56.5%) | 46 (44.2%) | 67 (69.8%) | | 95 (54.3%) | 13 (65%) | |
| Black | 70 (35%) | 48 (46.2%) | 22 (22.9%) | | 65 (37.1%) | 5 (25%) | |
| Other** | 17 (8.5%) | 10 (9.6%) | 7 (7.3%) | | 15 (8.6%) | 2 (10%) | |
| Ethnicity | | | | 0.70 | | | 0.14 |
| Hispanic or Latino | 60 (30%) | 30 (28.9%) | 30 (31.3%) | | 53 (30.3%) | 6 (30%) | |
| Non-Hispanic or Latino | 139 (69.5%) | 74 (71.1%) | 65 (67.7%) | | 122 (69.7%) | 13 (65%) | |
| Unknown | 1 (0.5%) | 0 | 1 (1%) | | 0 | 1 (5%) | |
| Diabetes | | | | 0.05 | | | 0.07 |
| No | 174 (87.9%) | 95 (92.2%) | 79 (83.2%) | | 155 (89.6%) | 15 (75%) | |
| Yes | 24 (12.1%) | 8 (7.7%) | 16 (16.8%) | | 18 (10.4%) | 5 (25%) | |
| Missing | 2 | 2 | | | 2 | | |
| % on medications to treat diabetes | 24 (12.1%) | 8 (7.7%) | 16 (16.8%) | 0.05 | 18 (10.4%) | 5 (25%) | 0.07 |
| History of AIDS | 56 (29%) | 25 (24.3%) | 31 (34.4%) | 0.12 | 49 (28.7%) | 7 (31.8%) | 0.76 |
| Waist circumference (cm) | 101.3 (15.9) | 97.3 (15.1) | 105.7 (15.6) | <0.01 | 100.4 (15.3) | 106.5 (16.7) | 0.10 |
| ALT (U/L) | 29.6 (20.1) | 22.8 (11.8) | 37.4 (24.3) | <0.01 | 28.1 (17.8) | 45.4 (31.9) | 0.03 |
| AST (U/L) | 25.4 (13.5) | 21.3 (7.9) | 30.0 (16.8) | <0.01 | 24.5 (12.2) | 34.6 (21.2) | 0.05 |
| Platelets($10^9$/L) | 235.5 (68.1) | 231.0 (68.6) | 240.7 (67.6) | 0.32 | 240.3 (68.2) | 206.8 (65.9) | 0.04 |
| Triglycerides (mg/dL) | 149.4 (115.0) | 123.4 (70.5) | 177.4 (144.0) | <0.01 | 150.7 (120.0) | 146.7 (81.0) | 0.89 |
| Fasting glucose (mg/dL) | 95.4 (50.7) | 81.9 (10.3) | 107.2 (67.5) | 0.16 | 92.3 (40.7) | 110.6 (91.4) | 0.68 |
| Insulin (μU/mL) | 25.3 (50.1) | 21.4 (37.6) | 29.0 (59.6) | 0.45 | 26.3 (53.6) | 20.0 (12.9) | 0.36 |
| Absolute CD4 | 744.1 (332.1) | 705.7 (336.6) | 786.1 (323.6) | 0.09 | 751.8 (326.8) | 674.5 (363.3) | 0.37 |
| Nadir CD4 | 227.3 (214.9) | 243.0 (223.6) | 209.8 (205.2) | 0.37 | 234.4 (219.6) | 199.1 (188.6) | 0.54 |
| % with Nadir CD4 <200 | 67 (33.8%) | 39 (36.8%) | 28 (30.4%) | 0.35 | 60 (35.1%) | 7 (31.8%) | 0.76 |
| Duration of HIV infection | 15.2 (9.9) | 14.0 (9.6) | 16.6 (10.1) | 0.07 | 14.9 (9.7) | 16.9 (11.8) | 0.38 |
| ART exposure (current/prior use) | 187 (93.5%) | 100 (96.2%) | 87 (90.6%) | 0.95 | 165 (94.3%) | 20 (100%) | 1.00 |
| % PI | 43 (21.5%) | 28 (26.9%) | 15 (15.6%) | 0.09 | 38 (21.7%) | 5 (25%) | 1.00 |
| % NNRTI | 32 (16%) | 12 (11.5%) | 20 (20.8%) | 0.05 | 28 (16%) | 4 (20%) | 0.77 |
| % INSTI | 151 (75.5%) | 82 (78.9%) | 69 (71.9%) | 0.70 | 133 (76%) | 18 (90%) | 0.79 |
| % NRTI | 178 (89%) | 96 (92.3%) | 82 (85.4%) | 0.74 | 157 (89.7%) | 21 (84%) | 1.00 |
| % Entry Inhibitors | 2 (1%) | 0 | 2 (2.1%) | 0.22 | 1 (0.6%) | 1 (5%) | 0.22 |
| For NRTI: | | | | | | | |
| % Abacavir | 12 (6.4%) | 9 (9%) | 3 (3.5%) | 0.12 | 10 (6.1%) | 2 (9.1%) | 0.64 |
| % Didanosine | 0 | 0 | 0 | | 0 | 0 | |
| % Emtricitabine | 159 (84.1%) | 84 (84%) | 75 (86.2%) | 0.67 | 141 (85.5%) | 18 (81.8%) | 0.75 |
| % Lamivudine | 19 (10.1%) | 12 (12%) | 7 (8.1%) | 0.37 | 16 (9.7%) | 3 (13.6%) | 0.47 |
| % Stavudine | 0 | 0 | 0 | | 0 | 0 | |
| % TDF | 36 (19.1%) | 24 (24%) | 12 (13.8%) | 0.08 | 29 (17.6%) | 7 (31.8%) | 0.15 |
| % TAF | 122 (64.6%) | 59 (59%) | 63 (72.4%) | 0.06 | 112 (67.9%) | 10 (45.5%) | 0.05 |
| % Zidovudine | 0 | 0 | 0 | | 0 | 0 | |

*(Continued)*

**Table 1.** (Continued)

| Variable | Total (N = 200) | No NAFLD (N = 104) | NAFLD (N = 96) | P-value | No CSF* (N = 175) | CSF (N = 20) | P-value |
|---|---|---|---|---|---|---|---|
| % Zalcitabine | 0 | 0 | 0 | | 0 | 0 | |
| HIV RNA (%) | | | | | | | |
| < 20 copies/ml | 31 (15.5%) | 15 (14.4%) | 16 (16.7%) | 0.66 | 26 (14.9%) | 4 (20%) | 0.52 |
| <200 copies/ml | 193 (96.5%) | 99 (95.2%) | 94 (97.9%) | 0.45 | 168 (96%) | 20 (100%) | 1.00 |

* Liver stiffness measurement (LSM) was available on 195 of 200 patients with ultrasound. CSF: Clinically significant fibrosis (defined as LSM ≥8.6 kPa)

** Other: combined other race categories with very few frequencies including Asians, Native Hawaiian, American Indians and Unknown races. AIDS: Acquired immunodeficiency syndrome, ART: Antiretroviral therapy, BMI: Body mass index, INSTI: Integrase strand transfer inhibitors, NNRTI: Non-nucleoside reverse transcriptase inhibitors, NRTI: Nucleoside reverse transcriptase inhibitors, PI: Protease inhibitors, TAF: Tenofovir Alafenamide, TDF: Tenofovir disoproxil fumarate

race, non-Hispanic ethnicity, higher triglycerides levels, insulin levels, and absolute and nadir $CD4^+$ T cell counts were associated with worse MCS, whereas older age, higher BMI, and larger waist circumference were associated with better MCS scores.

On multivariate analysis Table 3, diabetes was the only independent variable negatively associated with PCS in PWH, whereas Hispanic ethnicity (positively) and nadir $CD4^+$ T cell counts (negatively) were independently associated with MCS in PWH.

## Characteristics of participants with HIV-NAFLD compared to those with primary NAFLD

There were distinct differences in the characteristics of participants with NAFLD between the two groups (S2 Table). Participants with HIV-NAFLD, compared to those with primary NAFLD, were less frequently cisgender females (18.4% vs 62%), White (71.3% vs 94.3%), and

**Table 2. Summary and individual scores on the SF-36 questionnaire in persons with HIV.**

| Variable (N = 200) Mean (SD) | Overall | No NAFLD (N = 104) | NAFLD (n = 96) | P-value | NAFLD without CSF (n = 82) | NAFLD with CSF (n = 12) | P-value |
|---|---|---|---|---|---|---|---|
| Physical component summary score | 47.7 (11.0) | 47.4 (11.6) | 48.1 (10.3) | 0.64 | 48.6 (10.1) | 44.1 (11.3) | 0.16 |
| • Physical Functioning | 79.7 (26.5) | 77.4 (29.5) | 82.3 (22.7) | 0.19 | 83.0 (22.0) | 74.6 (27.6) | 0.23 |
| • Role limitations due to physical health | 75.4 (37.8) | 75.2 (37.9) | 75.5 (37.9) | 0.96 | 75.3 (38.4) | 75.0 (38.4) | 0.98 |
| • Pain | 70.8 (27.4) | 70.0 (26.5) | 71.7 (28.5) | 0.67 | 72.5 (27.7) | 66.1 (33.6) | 0.47 |
| • General health | 69.4 (23.6) | 69.5 (23.5) | 69.2 (23.8) | 0.92 | 69.8 (23.5) | 61.7 (26.0) | 0.27 |
| Mental component summary score | 50.3 (12.2) | 49.7 (12.0) | 50.9 (12.4) | 0.48 | 50.4 (12.4) | 53.0 (12.9) | 0.49 |
| • Emotional well-being | 75.0 (21.6) | 74.7 (19.7) | 75.2 (23.7) | 0.87 | 74.3 (23.0) | 77.7 (28.6) | 0.65 |
| • Role limitations due to emotional problems | 78.2 (37.7) | 75.0 (38.8) | 81.6 (35.5) | 0.21 | 80.5 (36.7) | 88.9 (29.6) | 0.45 |
| • Energy/fatigue | 63.6 (25.2) | 61.4 (25.2) | 65.9 (25.1) | 0.21 | 66.2 (24.0) | 59.6 (31.2) | 0.40 |
| • Social functioning | 79.0 (27.0) | 77.9 (25.6) | 80.2 (28.6) | 0.55 | 79.6 (28.8) | 81.3 (29.9) | 0.85 |

CSF: Clinically significant fibrosis defined as LSM ≥8.6 kPa.

**Table 3. Factors associated with health-related quality of life in persons with HIV based on multivariate analysis.**

| Variable | PCS (Step wise regression model) | | | Variable | MCS (Step wise regression) | | |
|---|---|---|---|---|---|---|---|
| | β estimate | SE | P-value | | β estimate | SE | P-value |
| NAFLD with CSF (ref = NAFLD without CSF) | -3.68 | 3.10 | 0.24 | NAFLD with CSF (ref = NAFLD without CSF) | -1.16 | 4.00 | 0.76 |
| Diabetes | | | | Hispanic | | | |
| Yes | -7.45 | 2.80 | 0.01 | Yes | 10.97 | 2.90 | <0.01 |
| No | Reference | | | No | Reference | | |
| | | | | Nadir CD4 | -0.02 | 0.01 | 0.01 |

CSF: clinically significant fibrosis

Multivariable model included controlled attenuation parameter (CAP), liver stiffness measurement (LSM) and all the variables that had a P-value of 0.2 or less in the univariate analysis

more frequently of Hispanic ethnicity (29.9% vs 1.3%). They had lower BMI (SD) [30.5 (5.7) vs 35.8 (7.4) kg/m$^2$] and lower frequency of obesity (44.8% vs 79.3%) and diabetes (17.4% vs 38%). Participants with HIV-NAFLD also had lower AST [30.2 (16.1) vs 35.0 (18.7) U/L] and lower frequency of clinically significant fibrosis (12.6% vs 45.2%) than those with primary NAFLD.

## Comparison of HRQOL in HIV-NAFLD and primary NAFLD

Overall, the two groups had poor physical and mental HRQOL as reflected by PCS and MCS ≤50. Participants with HIV-NAFLD had better physical quality of life and reported less fatigue compared to participants with primary NAFLD Table 4.

Participants with HIV-NAFLD also had significantly better domain scores for physical functioning [82.1 (22.7) vs 67.5 (28.7)], role limitations due to physical health [74.4 (38.3) vs 57.9 (43.4) vs], pain [70.8 (27.8) vs 58.1 (26.0) vs], general health [67.8 (24.1) vs 51.0 (23.1)], and better PCS [47.8 (10.2) vs 40.2 (12.0)]. Compared to primary NAFLD, participants with HIV-NAFLD reported significantly better energy and fatigue [64.5 (25.0) vs 42.5 (23.0)], with trends toward worse emotional well-being, role limitations due to emotional problems, social functioning and MCS.

## Factors associated with HRQOL in all participants with NAFLD (HIV and primary)

In a univariate analysis (S3 Table), variables associated with better PCS included HIV NAFLD, male sex, "Other" race, Hispanic ethnicity, and higher platelet levels, whereas

**Table 4. Comparison of health-related quality of life in HIV-NAFLD and primary NAFLD.**

| Variable Mean (SD) | HIV-NAFLD (n = 87) | Primary NAFLD (n = 474) | P-value |
|---|---|---|---|
| Physical component summary score | 47.8 (10.2) | 40.2 (12.0) | <0.01 |
| Physical functioning | 82.1 (22.7) | 67.5 (28.7) | <0.01 |
| Role limitations due to physical health | 74.4 (38.3) | 57.9 (43.4) | <0.01 |
| Pain | 70.8 (27.8) | 58.1 (26.0) | <0.01 |
| General health | 67.8 (24.1) | 51.0 (23.1) | <0.01 |
| Mental component summary score | 50.5 (12.6) | 48.2 (11.0) | 0.07 |
| Emotional well-being | 74.8 (24.3) | 70.7 (19.5) | 0.14 |
| Role limitations due to emotional problems | 80.8 (35.8) | 74.1 (38.7) | 0.13 |
| Energy/fatigue | 64.5 (25.0) | 42.5 (23.0) | <0.01 |
| Social functioning | 79.5 (28.2) | 74.3 (27.9) | 0.11 |

**Table 5. Factors associated with physical and mental components of health-related quality of life in patients with NAFLD in multivariate analysis.**

| Variable | PCS | | | MCS | | |
|---|---|---|---|---|---|---|
| | β estimate | SE | P-value | β estimate | SE | P-value |
| HIV-NAFLD (ref = General NAFLD) | 2.99 | 1.67 | 0.08 | -2.01 | 1.69 | 0.23 |
| CSF (ref = No CSF) | -2.85 | 1.01 | 0.01 | -0.44 | 1.02 | 0.66 |
| Sex | | | | | | |
| Female | Reference | | | Reference | | |
| Male | 3.21 | 0.99 | <0.01 | 3.49 | 1.00 | <0.01 |
| Race | | | | | | |
| White | Reference | | | Reference | | |
| Black | -0.55 | 2.46 | 0.82 | -1.75 | 2.44 | 0.47 |
| Other | 5.90 | 2.24 | 0.01 | 0.61 | 2.25 | 0.79 |
| Ethnicity | | | | | | |
| Non-Hispanic or Latino | Reference | | | Reference | | |
| Hispanic or Latino | -1.88 | 2.24 | 0.40 | 9.40 | 2.26 | <0.01 |
| Refused/Unknown | -1.63 | 1.11 | 0.14 | -0.56 | 1.12 | 0.62 |
| BMI | -0.29 | 0.07 | <0.01 | -0.04 | 0.07 | 0.59 |
| Diabetes | | | | | | |
| Yes | -4.76 | 1.01 | <0.01 | -0.50 | 1.01 | 0.63 |
| No | Reference | | | Reference | | |

CSF: Clinically significant fibrosis (defined as LSM $\geq$ 8.6 kPa)

clinically significant fibrosis, older age, higher BMI, diabetes, and higher triglyceride and glucose levels were associated with worse PCS. After adjustment for significant covariates in the multivariate analysis Table 5, there was no significant difference in PCS between participants with HIV-NAFLD compared to those with primary NAFLD. Only male sex and "Other" race were independently associated with better PCS, whereas clinically significant fibrosis and presence of diabetes were associated with worse PCS.

Factors associated with better MCS in univariate analysis were older age, male sex and Hispanic ethnicity whereas higher platelets, triglycerides, glucose and insulin levels were associated with worse MCS (**S3 Table**). Of these variables, only male sex and Hispanic ethnicity were independently associated with better MCS in the multivariate analysis Table 5.

## Discussion

With the availability of effective therapies for viral hepatitis, NAFLD has emerged as the leading cause of liver disease in PWH [16]. HIV-NAFLD was not associated with impairment in physical or mental HRQOL in this study. Importantly, this multicenter cohort of persons with HIV mono-infection was comprised predominantly of persons who are on ART and had adequate viral suppression. In this setting, diabetes, Hispanic ethnicity and nadir CD4[+] T cell counts, but not NAFLD or clinically significant fibrosis, were associated with impaired HRQOL in PWH. Further, after adjustment for significant covariates, there was no difference in HRQOL between HIV and primary NAFLD. Clinically significant fibrosis, diabetes and demographic variables, but not HIV serostatus, were independently associated with HRQOL in NAFLD (HIV and primary).

There are sparse data examining the association of HIV-NAFLD with HRQOL. While primary NAFLD was shown to be associated with worse physical HRQOL compared to the general population [18–20], we did not detect a significant association for ultrasound-diagnosed

HIV-NAFLD with HRQOL in this study. A recent single center study from Germany evaluated HRQOL in 245 PWH using a CAP of $\geq$ 275 dB/m to define fatty liver and the European Quality-of-Life 5-Dimension 5-Level questionnaire [36]. The study reported 35% prevalence of fatty liver (27.2% due to NAFLD and remaining due alcohol) with a mean BMI in that cohort of 25.1 kg/m$^2$ and 29.4% of participants with HIV RNA above the chosen 50 copies/ml threshold of suppression. While HRQOL in that study was lower in PWH and fatty liver (due to combined NAFLD and alcohol) than PWH without fatty liver, fatty liver was not independently associated with HRQOL on multivariate analysis, whereas unemployment and waist circumference were. That fatty liver was not independently associated with HRQOL in that study is consistent with our findings. In a follow up study, the same group from Germany used an HIV-specific tool (MOS-HIV survey) to assess HRQOL in PWH [37]. In addition to confirming the importance of metabolic factors, lower socioeconomic status and presence of significant fibrosis were also noted to negatively affect the HRQOL in PLWH.

Diabetes was the only independent factor strongly and negatively associated with physical HRQOL in PWH in our study. Diabetes is also independently associated with worse physical HRQOL in studies of primary NAFLD [20, 22, 38, 39].

In studies of primary NAFLD, NASH, advanced fibrosis ($\geq$F3) or cirrhosis (F4) but not less severe stages of fibrosis were shown to be associated worse physical HRQOL [20, 22, 38, 39]. In this study, the presence of clinically significant fibrosis ($\geq$F2 by LSM) in HIV-NAFLD was not significantly associated with the mental or physical HRQOL in PWH, similar to the previously mentioned study [36]. Since only 10% of PWH in our study had clinically significant fibrosis, and even a smaller proportion of them with advanced fibrosis, we were unable to evaluate the association of advanced fibrosis or cirrhosis with HRQOL. It is possible the low prevalence of advanced fibrosis may explain the lack of association between NAFLD and HRQOL in PWH in this cohort.

Participants with HIV-NAFLD have distinctly different demographic, metabolic and laboratory characteristics than those with primary NAFLD. Participants with HIV-NAFLD were less frequently cisgender females, White, and more frequently of Hispanic ethnicity than those with primary NAFLD. They also had lower BMI and lower frequency of obesity, diabetes and clinically significant fibrosis. Yet, after adjustment for significant covariates, there was no difference in HRQOL between participants with HIV and primary NAFLD.

Despite concerns about increased metabolic complications PWH experience from ART, HIV serostatus was not associated with components of physical or mental HRQOL in the analysis of all persons with NAFLD (HIV and primary NAFLD) in this study. Rather, as in studies of primary NAFLD [20, 22], clinically significant fibrosis and diabetes were the main factors independently associated with worse physical HRQOL.

In PWH, we did not observe an independent association between gender and HRQOL. However, in patients with NAFLD (with and without HIV), male sex was independently associated with better HRQOL as reflected by better PCS and MCS. Complex demographic, cultural, racial and ethnic factors interact to influence the support an individual receives to help cope with a health condition, which in turn may affect the individual's sense of wellbeing. In this study, male sex and "Other" race, but not HIV serostatus (HIV NAFLD), were the only independent factors associated with better physical HRQOL in patients with NAFLD, whereas male sex and Hispanic ethnicity were associated with better mental HRQOL in patients with NAFLD. The role of socioeconomic and demographic factors in influencing mental HRQOL has been shown in studies of patients with NAFLD, HIV and other chronic conditions such as lupus [22, 38, 40–42].

As reported in our study, PWH have very high prevalence of NAFLD ranging from 35–59%, and those with HIV-NAFLD have prevalence of significant fibrosis ranging from 7–20%.

[15, 36, 43]. Thus, these data suggest that screening for high risk NAFLD with hepatic fibrosis is warranted in PWH.

This study has several strengths. Participants were prospectively enrolled and underwent detailed systematic phenotyping. PWH were from diverse demographic, racial and ethnic backgrounds, and the study's two cohorts were contemporaneously prospectively enrolled over the same time period. The study also has a few limitations. Nearly all PWH were on ART and had achieved adequate HIV-1 viral suppression; therefore, it is unknown if our findings apply to PWH not on ART or who are not adequately suppressed. We did not collect data on socioeconomic status, body composition or physical activity, factors that may influence HRQOL.

In summary, diabetes, non-Hispanic ethnicity, and nadir $CD4^+$ T cell counts, but not NAFLD or clinically significant fibrosis, were associated with impaired HRQOL in PWH on ART achieving adequate HIV viral suppression. In a combined cohort of NAFLD that included persons with HIV-NAFLD on ART and primary NAFLD, clinically significant fibrosis, diabetes, and demographic factors, but not HIV serostatus, were associated with decreased HRQOL.

## Supporting information

**S1 Table. Factors associated with HRQOL in PWH on univariate analysis.**
(DOCX)

**S2 Table. Comparison of subjects' characteristics between primary NAFLD and HIV NAFLD.**
(DOCX)

**S3 Table. Factors associated with physical and mental components of HRQOL in patients with NAFLD in univariate analysis.**
(DOCX)

## Author Contributions

**Conceptualization:** Samer Gawrieh, Kathleen E. Corey, Jordan E. Lake, Niharika Samala, Archita P. Desai, Naga Chalasani.

**Data curation:** Samer Gawrieh, Kathleen E. Corey, Jordan E. Lake, Niharika Samala, Paula Debroy, Julia A. Sjoquist, Montreca Robison, Mark Tann, Fatih Akisik, Surya S. Bhamidipalli, Chandan K. Saha, Kimon Zachary, Gregory K. Robbins, Samir K. Gupta.

**Formal analysis:** Samer Gawrieh, Kathleen E. Corey, Jordan E. Lake, Niharika Samala, Archita P. Desai, Paula Debroy, Surya S. Bhamidipalli, Chandan K. Saha, Kimon Zachary, Gregory K. Robbins, Samir K. Gupta, Naga Chalasani.

**Writing – original draft:** Samer Gawrieh, Kathleen E. Corey, Jordan E. Lake, Niharika Samala, Archita P. Desai, Naga Chalasani.

**Writing – review & editing:** Samer Gawrieh, Kathleen E. Corey, Jordan E. Lake, Niharika Samala, Paula Debroy, Surya S. Bhamidipalli, Chandan K. Saha, Kimon Zachary, Gregory K. Robbins, Samir K. Gupta, Raymond T. Chung, Naga Chalasani.

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
