## [Decision Letter · Decision Letter 0]

24 Oct 2022

PONE-D-22-24135Non-alcoholic Fatty Liver Disease is Not Associated with Impairment in Health-related Quality of Life in Virally Suppressed Persons with Human Immune Deficiency VirusPLOS ONE

Dear Dr. Gawrieh,

Thank you for submitting your manuscript to PLOS ONE. After careful consideration, we feel that it has merit but does not fully meet PLOS ONE’s publication criteria as it currently stands. Therefore, we invite you to submit a revised version of the manuscript that addresses the points raised during the review process. As you can see, both reviewers appreciated your work and only relatively minor changes are needed.

We look forward to receiving your revised manuscript.

Kind regards,

Pavel Strnad

Academic Editor

PLOS ONE

Journal Requirements:

"Dr. Gawrieh consulting: TransMedics, Pfizer. Research grant support: Cirius, Galmed, Viking and Zydus. Dr. Corey serves on the scientific advisory board for Theratechnologies, Novo Nordisk and BMS and has received grant funding from Boehringer-Ingelheim, BMS and Novartis, Dr. Lake receives research support from Gilead Sciences and Zydus. In the past 12 months she has received research support from Pfizer, CytoDyn, and Oncoimmune, and has served as a consultant to Merck and Theratechnologies, Dr. Samala, Dr. Desai, Dr. Debroy, Ms.Sjoquist, Ms. Robison, Ms. Bhamidipalli, Dr. Saha, Dr. Zachary, Dr. Akisik, and Dr. Tann have nothing to disclose. Dr. Robbin has served as a consultant for SEED , Dr. Gupta reports consultancy/advisory fees from Gilead Sciences, Inc. and ViiV Healthcare and research support from the NIH, Indiana University School of Medicine, and ViiV HealthCare, Dr. Chung has received research grant support (to institution) from Boehringer Ingelheim, BMS, Roche, Gilead, Janssen, and GSK , Dr. Chalasani has ongoing research support from Eli Lilly, Galectin Therapeutics, Intercept, and Exact Sciences, In the past 12 months, he has received consulting fees from Abbvie, Madrigal, Nusirt, Allergan, Siemens, Genentech, Zydus, La Jolla, Axcella, Foresite Labs, and Galectin Therapeutics."

4. Please include a copy of Table 4 which you refer to in your text on page 14.

Reviewers' comments:

Reviewer's Responses to Questions

**Comments to the Author**

1. Is the manuscript technically sound, and do the data support the conclusions?

Reviewer #1: Yes

Reviewer #2: Yes

2. Has the statistical analysis been performed appropriately and rigorously? 

Reviewer #1: Yes

Reviewer #2: I Don't Know

3. Have the authors made all data underlying the findings in their manuscript fully available?

Reviewer #1: Yes

Reviewer #2: Yes

4. Is the manuscript presented in an intelligible fashion and written in standard English?

Reviewer #1: Yes

Reviewer #2: Yes

5. Review Comments to the Author

Reviewer #1: This analysis assesses quality of life in NAFLD with a generic tool and compared HIV-infected and non-infected patients. The strength is the large cohort and the presence of a control group. Limitations arise from the use of a single non-liver disease and non-HIV specific tool. Overall, this is an important study and the information will guide the field. I have included comments below:

The abstract needs to be checked – in the first sentence liver “disease” is missing

The prevalence of hepatic steatosis exceeds the estimates in the general population by far! This is an interesting observation that has been captured in other cohorts, e.g. PMID: 35996039. Do the authors consider screening for NAFLD in HIV ? A section in the discussion would be welcome. The same study reported on an HIV-specific tool and observed impact of significant fibrosis on QoL – this could be added in the discussion (PMID: 35996039)

It could be mentioned that alcohol assessment (ADUIT >8) leaves room for harmful alcohol consumption if not assessed clinically.

QoL is strongly impacted by sex with women reporting lower levels of QoL in all analysis. I could not see lower OoL in this analysis (table 2) – clearly here a male dominance and transgender aspect could be the reason for this. Then again male sex was associated with higher levels in supp. Table 3. Please comment.

Please indicate when the SF-36 was completed in relation to the diagnostic test for NAFLD. At the same time? Up to 6 or 12 month ?

In the literature there have been reports of TAF impacting obesity – can you provide data on the ART backbone?

Reviewer #2: The authors describes very well Quality of life related outcomes in PLWH.

However some additional data are required:

-In table 1 must be added the antiretroviral drug classes that PLWH are taking and the duration of HIV infection

-nadir CD4 is significantly associated with HRQOL ; how many patients were affected by severe opportunistic infections during their clinical history?

-no data on therapy for diabetes are included ; it could be better to have data on the percentages of patients on therapy because these data could impact on HRQOL

6. PLOS authors have the option to publish the peer review history of their article (what does this mean?). If published, this will include your full peer review and any attached files.

Reviewer #1: **Yes: **Jörn Schattenberg

Reviewer #2: No

---

## [Author Response · Author response to Decision Letter 0]

7 Dec 2022

December 7, 2022

Dr. Pavel Strnad

Academic Editor

PLOS One

Manuscript PONE-D-22-24135: “Non-alcoholic Fatty Liver Disease is Not Associated with Impairment in Health-related Quality of Life in Virally Suppressed Persons with Human Immune Deficiency Virus”

Dear Dr. Strnad,

We would like to thank you and the Reviewers for your insightful and constructive comments that have helped us improve and strengthen our manuscript. Based on your comments, we have modified the manuscript. Below, we have addressed all specific comments individually and made the corresponding changes to the draft.

Competing interests: Dr. Gawrieh consulting: TransMedics, Pfizer. Research grant support: Viking, Zydus, and Sonic Incytes. Dr. Corey serves on the scientific advisory board for Theratechnologies, Novo Nordisk and BMS and has received grant funding from Boehringer-Ingelheim, BMS and Novartis, Dr. Lake receives research support from Gilead Sciences and Zydus. In the past 12 months she has received research support from Pfizer, CytoDyn, and Oncoimmune, and has served as a consultant to Merck and Theratechnologies, Dr. Samala, Dr. Desai, Dr. Debroy, Ms. Sjoquist, Ms. Robison, Ms. Bhamidipalli, Dr. Saha, Dr. Zachary, and Dr. Tann have nothing to disclose , Dr. Akisik…., Dr. Robbin has served as a consultant for SEED , Dr. Gupta reports consultancy/advisory fees from Gilead Sciences, Inc. and ViiV Healthcare and research support from the NIH, Indiana University School of Medicine, and ViiV HealthCare, Dr. Chung has received research grant support (to institution) from Boehringer Ingelheim, BMS, Roche, Gilead, Janssen, and GSK , Dr. Chalasani has ongoing research support from Eli Lilly, Galectin Therapeutics, Intercept, and Exact Sciences, In the past 12 months, he has received consulting fees from Abbvie, Madrigal, Nusirt, Allergan, Siemens, Genentech, Zydus, La Jolla, Axcella, Foresite Labs, and Galectin Therapeutics. 

This does not alter our adherence to PLOS ONE policies on sharing data and materials

We hope you will consider our revised paper for publication in PLOS One.

Sincerely,

Samer Gawrieh, MD on behalf of all the authors 

Indiana University School of Medicine

Journal Comments

Journal Requirements:

Authors response: Done. We have reformatted our manuscript according to PLOS ONE's style requirements.

Authors response: Done. These statements are now included in the methods: 

“ Each participant provided a signed informed consent. The study protocol was approved by each site’s Institutional Review Board (IRB).”

AND 

“Indiana University IRB approved the protocol. Each participant provided a signed informed consent.”

"Dr. Gawrieh consulting: TransMedics, Pfizer. Research grant support: Cirius, Galmed, Viking and Zydus. Dr. Corey serves on the scientific advisory board for Theratechnologies, Novo Nordisk and BMS and has received grant funding from Boehringer-Ingelheim, BMS and Novartis, Dr. Lake receives research support from Gilead Sciences and Zydus. In the past 12 months she has received research support from Pfizer, CytoDyn, and Oncoimmune, and has served as a consultant to Merck and Theratechnologies, Dr. Samala, Dr. Desai, Dr. Debroy, Ms.Sjoquist, Ms. Robison, Ms. Bhamidipalli, Dr. Saha, Dr. Zachary, Dr. Akisik, and Dr. Tann have nothing to disclose. Dr. Robbin has served as a consultant for SEED , Dr. Gupta reports consultancy/advisory fees from Gilead Sciences, Inc. and ViiV Healthcare and research support from the NIH, Indiana University School of Medicine, and ViiV HealthCare, Dr. Chung has received research grant support (to institution) from Boehringer Ingelheim, BMS, Roche, Gilead, Janssen, and GSK , Dr. Chalasani has ongoing research support from Eli Lilly, Galectin Therapeutics, Intercept, and Exact Sciences, In the past 12 months, he has received consulting fees from Abbvie, Madrigal, Nusirt, Allergan, Siemens, Genentech, Zydus, La Jolla, Axcella, Foresite Labs, and Galectin Therapeutics."

Authors response: This statement has now been added to the competing interests section:

“This does not alter our adherence to PLOS ONE policies on sharing data and materials.”

Authors response: Done. Thank you.

4. Please include a copy of Table 4 which you refer to in your text on page 14.

Authors response: Sorry about that. We have now included Table 4.

Authors response: We have now included matching captions for the supporting information in the text and at the end of the manuscript.

Authors response: Reference list is correct. To our knowledge, no reference has been retracted. We have added the reference suggested by Reviewer #1.

Reviewer #1: 

The abstract needs to be checked – in the first sentence liver “disease” is missing

Authors response: Thank you for catching this. We have now checked the abstract again. We also added the word “disease” to the first sentence

The prevalence of hepatic steatosis exceeds the estimates in the general population by far! This is an interesting observation that has been captured in other cohorts, e.g. PMID: 35996039. Do the authors consider screening for NAFLD in HIV ? A section in the discussion would be welcome. 

Authors response: Thank you for this important comment. We have now added the following section in the discussion to highlights the need for screening for NAFLD in PWH: 

As reported in our study, PWH have very high prevalence of NAFLD ranging from 35-59%, and those with HIV-NAFLD have prevalence of significant fibrosis ranging from 7-20% (15, 36, 43). Thus, these data suggest that screening for high risk NAFLD with hepatic fibrosis is warranted in PWH.

The same study reported on an HIV-specific tool and observed impact of significant fibrosis on QoL – this could be added in the discussion (PMID: 35996039)

Authors response: Thank you for this note. We have now added the following section in the discussion to highlights this study: 

In a follow up study, the same group from Germany used an HIV-specific tool (MOS-HIV survey) to assess HRQOL in PWH (37). In addition to confirming the importance of metabolic factors, lower socioeconomic status and presence of significant fibrosis were also noted to negatively affect the HRQOL in PLWH. 

It could be mentioned that alcohol assessment (ADUIT >8) leaves room for harmful alcohol consumption if not assessed clinically.

Authors response: Thank you for this comment. At your suggestion, we have now added the following paragraph to the discussion.

In this study, we applied an alcohol use questionnaire, AUDIT, to survey participants for excessive and harmful drinking. While this may not be feasible in clinical practice, obtaining alcohol consumption history during clinical encounters is a feasible and alternative way to screen for harmful alcohol consumption.

QoL is strongly impacted by sex with women reporting lower levels of QoL in all analysis. I could not see lower OoL in this analysis (table 2) – clearly here a male dominance and transgender aspect could be the reason for this. Then again male sex was associated with higher levels in supp. Table 3. Please comment.

Authors response: We thank the Reviewer for this comment. Please allow us to clarify.

When we look at HRQOL specifically in PWH, we see numerical but non-significant differences in PCS and MCS between women and men (Supplemental table 1). Indeed, after adjusting for covariates, gender was not an independent factor influencing PCS or MCS in PWH (Table 3 multivariable analysis of HRQOL in PWH). 

When we look at ALL patients with NAFLD (with and without HIV), data shown in supplementary Table 3 (Univariate analysis) and main Table 5 (multivariate analysis), women had significantly lower PCS and MCS than men.

To clarify this, we have added the following paragraph in the discussion

In PWH, we did not observe an independent association between gender and HRQOL. However, in patients with NAFLD (with and without HIV), male sex was independently associated with better HRQOL as reflected by better PCS and MCS.

Please indicate when the SF-36 was completed in relation to the diagnostic test for NAFLD. At the same time? Up to 6 or 12 month ?

Authors response: The SF-36 was completed the same day of the diagnostic testing for NAFLD (ultrasound and Fibroscan). We have now clarified this in the methods and added the following sentence

At the time of enrollment (same day of the diagnostic testing for NAFLD), participants also completed the 36-item Short Form (SF-36) Health Survey version 1.0. 

In the literature there have been reports of TAF impacting obesity – can you provide data on the ART backbone?

Authors response: Thank you for this suggestion. We have now added the detailed ART data to Table 1. The Reviewer is correct, there was a trend for PWH with NAFLD to have higher exposure to TAF. We have added the following sentence to the results to summarize additional the ART data:

PWH with HIV NAFLD, compared to PWH without HIV NAFLD tended to have longer duration of HIV infection (16.6 (10.1) vs 14.0 (9.6) years, P =0.07), had more exposure to non-nucleoside reverse transcriptase inhibitors (20.8% vs 11.5%, P = 0.05) and Tenofovir Alafenamide (72.4% vs 59%, P = 0.06). 

Reviewer #2: 

-In table 1 must be added the antiretroviral drug classes that PLWH are taking and the duration of HIV infection

Authors response: Thank you for this suggestion. We have now added these data to Table 1 and updated the results sections to reflect that by adding this paragraph:

PWH with HIV NAFLD, compared to PWH without HIV NAFLD tended to have longer duration of HIV infection (16.6 (10.1) vs 14.0 (9.6) years, P =0.07), had more exposure to non-nucleoside reverse transcriptase inhibitors (20.8% vs 11.5%, P = 0.05) and Tenofovir Alafenamide 72.4% vs 59%, P = 0.06).

-nadir CD4 is significantly associated with HRQOL ; how many patients were affected by severe opportunistic infections during their clinical history?

Authors response: Thank you for this suggestion. We have now added the proportion of participants with nadir CD4 <200 and history of AIDS data to table 1 and added the following paragraph to the results: 

There were no differences between the two groups in age, proportion with history of acquired immunodeficiency syndrome, absolute or nadir CD4+ T cell counts, proportion with nadir CD4 <200, ART exposure, or HIV-1 RNA suppression levels Table 1. 

-no data on therapy for diabetes are included ; it could be better to have data on the percentages of patients on therapy because these data could impact on HRQOL

Authors response: Thank you for this suggestion. All PWH who had diabetes were on hypoglycemic agents with a trend for PWH with NAFLD to be on these more than PWH without diabetes. We have now added the proportion of participants on diabetes medications to Table 1 and to the results section: 

PWH with HIV NAFLD, compared to PWH without HIV NAFLD, …… also tended to have higher frequency of type 2 diabetes (16.8% vs 7.7%, p=0.05) and use of hypoglycemic agents (16.8% vs 7.7%, p=0.05).

---

## [Decision Letter · Decision Letter 1]

14 Dec 2022

Non-alcoholic Fatty Liver Disease is Not Associated with Impairment in Health-related Quality of Life in Virally Suppressed Persons with Human Immune Deficiency Virus

PONE-D-22-24135R1

Dear Dr. Gawrieh,

We’re pleased to inform you that your manuscript has been judged scientifically suitable for publication and will be formally accepted for publication once it meets all outstanding technical requirements.

Kind regards,

Pavel Strnad

Academic Editor

PLOS ONE

Additional Editor Comments (optional):

Thank you for giving me the opportunity to handle this interesting manuscript!

Reviewers' comments:

Reviewer's Responses to Questions

**Comments to the Author**

1. If the authors have adequately addressed your comments raised in a previous round of review and you feel that this manuscript is now acceptable for publication, you may indicate that here to bypass the “Comments to the Author” section, enter your conflict of interest statement in the “Confidential to Editor” section, and submit your "Accept" recommendation.

Reviewer #1: All comments have been addressed

Reviewer #2: All comments have been addressed

2. Is the manuscript technically sound, and do the data support the conclusions?

Reviewer #1: Yes

Reviewer #2: Yes

3. Has the statistical analysis been performed appropriately and rigorously? 

Reviewer #1: Yes

Reviewer #2: I Don't Know

4. Have the authors made all data underlying the findings in their manuscript fully available?

Reviewer #1: Yes

Reviewer #2: Yes

5. Is the manuscript presented in an intelligible fashion and written in standard English?

Reviewer #1: Yes

Reviewer #2: Yes

6. Review Comments to the Author

Reviewer #1: Nothing to add - thorough analysis - in their revision, my previous comments have been addressed.

Reviewer #2: (No Response)

7. PLOS authors have the option to publish the peer review history of their article (what does this mean?). If published, this will include your full peer review and any attached files.

Reviewer #1: **Yes: **Jörn M. Schattenberg M.D.

Reviewer #2: No

---

## [Editor Report · Acceptance letter]

1 Feb 2023

PONE-D-22-24135R1 

Non-alcoholic fatty liver disease is not associated with impairment in health-related quality of life in virally suppressed persons with human immune deficiency virus 

Dear Dr. Gawrieh:

I'm pleased to inform you that your manuscript has been deemed suitable for publication in PLOS ONE. Congratulations! Your manuscript is now with our production department. 

Kind regards, 

on behalf of

Dr. Pavel Strnad 

Academic Editor

PLOS ONE